# Trends and Associated Factors of Dietary Knowledge among Chinese Older Residents: Results from the China Health and Nutrition Survey 2004–2015

**DOI:** 10.3390/ijerph17218029

**Published:** 2020-10-31

**Authors:** Shizhen Wang, Ying Yang, Runhu Hu, Hongfei Long, Ni Wang, Quan Wang, Zongfu Mao

**Affiliations:** 1School of Health Sciences, Wuhan University, Wuhan 430071, China; wangshzh@whu.edu.cn (S.W.); hurunhu@whu.edu.cn (R.H.); 2018203050035@whu.edu.cn (N.W.); zfmao@whu.edu.cn (Z.M.); 2Global Health Institute, Wuhan University, Wuhan 430071, China; 3Dong Fureng Economic & Social Development School, Wuhan University, Wuhan 430071, China; hongfei0411@sina.com

**Keywords:** dietary knowledge, Chinese elderly, trend, nutrition

## Abstract

Promoting a healthy diet of the elderly is an important task in the current “Healthy China Action”. This study aimed to describe the changing trends of the dietary knowledge elderly Chinese during 2004–2015 and to examine the associated factors of dietary knowledge. Elderly people aged ≥60 years were included as study subjects from the China Health and Nutrition Survey 2004–2015. A total of 15,607 samples were involved in the analysis. The correct rate of dietary knowledge items followed upward trends over time, except for two items regarding physical activity intensity (Question 11, Cochran-Armitage *χ*^2^ = 20.05, *p* < 0.001) and healthy weight (Question 12, Cochran-Armitage *χ*^2^ = 43.93, *p* < 0.001). Four of the twelve dietary knowledge items consistently followed the lowest correct rate between 2006 and 2015, regarding physical activity intensity (Question 11, 24.5%−25.8%), staple food consumption (Question 5, 36.6%−41.5%), animal product consumption (Question 6, 45.8%−59.5%), and fatty meat and animal fat consumption (Question 7, 63.6%−64.9%). Participants who had a lower educational level or lived in rural areas or western regions, did not know about the Chinese Food Pagoda (CFP) or Dietary Guidelines for Chinese Residents (DGCR), and did not proactively look for nutrition knowledge were less likely to have adequate dietary knowledge literacy. Targeted interventions should be developed to promote dietary knowledge level of the elderly.

## 1. Introduction

Globally, the share of the population aged 65 years or older increased from 6% in 1990 to 9% in 2019, and is predicted to reach 16% by 2050 [1]. In China, the population aged 60 years and older has reached 253 million, which accounted for 18.1% of the total population by the end of 2019, and this number is predicted to reach 34.9% by 2050 [2,3]. Accordingly, the health management and health promotion of the elderly are becoming an important topic in current China.

Dietary habits are closely related to the occurrence and mortality of chronic non-communicable diseases [4,5,6]. According to the Global Burden of Disease Study 2017, dietary risks were responsible for 11 million deaths (22% of all deaths among adults) and 255 million Disability-Adjusted Life Years (DALYs) (15% of all DALYs among adults) [7]. For elderly people, a balanced diet and optimum nutrition were conducive to their fitness, which can enhance their immunity to resist the invasion of chronic diseases [8]. However, previous studies found that elderly Chinese are facing the problems of unbalanced and unhealthy diet, such as high nutritional risk, insufficient intake of retinol, thiamine, riboflavin, vitamin C, and calcium [9,10].

Dietary knowledge is the basis of individual diet-related behavior change [11]. Previous studies suggested that dietary knowledge was significantly associated with the elderly’s dietary attitude and behaviors [12,13]. Spronk et al.’s [14] systematic review reported a significant positive correlation between higher nutrition knowledge and the adequate intake of fruit and vegetables. Kristine et al. [15] further indicated that nutrition knowledge was a modifiable determinant of diet quality, which may mediate the effects of socio-demographic characteristics on diet quality. Proper and adequate dietary knowledge can help the elderly to choose a healthy diet to promote their physical health [16]. However, elderly people are confronting a low level of dietary knowledge in China, and less than half of them know basic knowledge like dietary guidelines [17]. Few studies focused on the changing trends of dietary knowledge of Chinese older adults [18,19]. Generally, knowledge is limited regarding the changing trend of dietary knowledge over time and the weak points of dietary knowledge among Chinese older adults.

Previous studies reported the influencing factors of dietary knowledge, including education level, place of residence, and income level, etc. [20,21] Elderly people with a higher education level, living in urban areas, and with higher income generally have better dietary knowledge [22,23]. Some controversial results were also reported regarding the associated factors of the dietary knowledge level. For example, Zhou et al. [24] and Yu et al. [25] found that the dietary knowledge of adults had no significant influence on overweight and obesity, possibly due to a lack of systematic dietary knowledge and inadequate guidance on overweight and obesity. While Laz et al. [26] suggested that an increase in nutrition knowledge may promote healthy weight control behaviors among low-income reproductive-age women. In the field of gender, research showed that females tended to have higher dietary knowledge than males [20,27,28]. However, there were some contrary findings that dietary knowledge was uncorrected with gender [21,29]. As for age, certain studies suggested that the age of older adults was negatively associated with dietary knowledge [20,30], while others indicated that there was no association between them [13,21].

In July 2019, the Chinese government implemented “Appropriate Diet” and “Elderly Health Promotion” as part of the “Healthy China Action (2019–2030)” [31]. The “National Nutrition Plan (2017–2030)” also emphasized the importance of promoting the popularization of nutrition knowledge [32]. Thus, to clarify the above issue, and to provide references for the development of targeted intervention towards dietary health of the elderly, we conducted this study. The purposes of this study were to (a) describe the changing trends of dietary knowledge of the Chinese elderly between 2004–2015, and (b) examine the associated factors of Chinese elderly’s dietary knowledge.

## 2. Materials and Methods

### 2.1. Data Sources

This study used data from the China Health and Nutrition Survey (CHNS). In China, the CHNS aimed to investigate the impact of social and economic transitions in Chinese society on residents’ overall health and nutrition status. This longitudinal survey adopted multistage and random cluster procedures, and was first carried out in 1989 covering eight provinces. By 2015, the CHNS covered fifteen provinces in China, including Liaoning, Shandong, Henan, Jiangsu, Hubei, Hunan, Guizhou, Guangxi, Heilongjiang, Beijing, Shanghai, Chongqing, Zhejiang, Yunnan, and Shanxi. CHNS collects data on socioeconomic factors, health indicators, diet, and nutritional status, and the content of each round is adjusted appropriately according to the social and economic development and the change of residents’ lifestyle. In 2004, the CHNS began to collect information regarding dietary nutrition knowledge using 12 items, and five more items were added in 2015. More details regarding the study design and sampling strategies are available at the CHNS website (http://www.cpc.unc.edu/projects/china) or from Popkin et al. [33].

### 2.2. Study Population

In this study, individuals aged 60 years and older were included as study subjects from the CHNS database. In the description of dietary knowledge trends, CHNS data in 2004, 2006, 2009, 2011, and 2015 were used. A total of 15,607 samples were involved after excluding these with incomplete information in the 12 dietary knowledges (Q1–Q12). In the analysis of associated factors, CHNS data in 2015 were used. After excluding these with missing information in socio-demographic characteristics and five additional dietary knowledges (Q13–Q17), 4150 samples were included. Figure 1 illustrates the flow chart of the sample selection.

### 2.3. Measures

#### 2.3.1. Demographic Information

Participants’ socio-demographic characteristics were collected, including age, gender, educational level, nationality, body mass index (BMI), places of residence, and geographical regions. Age was divided into 60–69, 70–79, and ≥80 years old. Educational level included below primary school, primary school, middle school, and high school or above. Nationality was dichotomized into the Han nationality and the minority nationality. BMI was divided into three categories (<20.0 kg/m^2^, 20.0–26.9 kg/m^2^, and ≥27.0 kg/m^2^) based on the BMI criteria for Chinese elderly from Chinese Nutrition Society (CNS), i.e., the elderly’s BMI is preferably not less than 20.0 kg/m^2^ and not more than 26.9 kg/m^2^ [34]. The place of residence included urban areas and rural areas. The geographical region was categorized into the eastern regions (Beijing, Shanghai, Jiangsu, Zhejiang, and Shandong), the central regions (Henan, Hubei, and Hunan), the western regions (Guangxi, Guizhou, Yunnan, Chongqing, and Shanxi), and the northeast regions (Liaoning and Heilongjiang) based on the economic zone division criteria from China’s National Bureau of Statistics [35].

In addition, we included two indicators related to dietary knowledge: knowing about the Chinese Food Pagoda (CFP) [36] or Dietary Guidelines for Chinese Residents (DGCR) [37] and proactively looking for nutrition knowledge. The first indicator was calculated based on the question “Do you know about the Chinese Food Pagoda or the Dietary Guidelines for Chinese Residents (yes/no)?” The second indicator was measured by the question “Do you proactively look for nutrition knowledge (yes/no)?”

#### 2.3.2. Dietary Knowledge

Seventeen statements related to dietary knowledge were used to measure the participants’ dietary knowledge, including 12 items in 2004–2015 and 5 additional items in 2015. Each statement was originally coded as “strongly disagree”, “disagree”, “neutral”, “agree”, and “strongly agree”. For nine positive items (Q1, Q3, Q5, Q7, Q8, Q9, Q10, Q13, and Q17), the response of “strongly agree” or “agree” was considered as the correct answer. For the other eight negative items (Q2, Q4, Q6, Q11, Q12, Q14, Q15, and Q16), the response of “strongly disagree” or “disagree” was considered as the correct answer [11,18].

Based on the 17 dietary knowledge items in 2015, we computed the participants’ dietary knowledge literacy. Each item with the correct answer was scored 1 point, otherwise 0. Individuals with an actual dietary knowledge score ≥80% of the full score were defined as having adequate dietary knowledge literacy [11,38], i.e., the total score of the 17 dietary knowledge ≥14 in this study. Cronbach’s alpha for the 17 dietary questions was 0.86 in the study.

### 2.4. Statistical Analysis

Data analysis was performed using IBM SPSS 23.0 for Windows (IBM Corporation, Armonk, NY, USA). A two-sided *p*-value < 0.05 was considered as statistically significant. Descriptive analysis was used to describe the condition of dietary knowledge in 2004–2015. The secular trends of dietary knowledge were tested using the Cochran-Armitage test. Chi-square test was applied to compare dietary knowledge literacy among participants with different demographic characteristics. Variables with *p*-value < 0.1 in chi-square tests were selected as independent variables for the regression model. Binary logistic regression analysis was used to examine the associated factors of dietary knowledge. The condition of dietary knowledge literacy (yes or no) was set as the dependent variable. All predictor variables were entered into the same model, and the model selection was automated.

### 2.5. Ethics Statements

Protocols, instruments, and the process used to obtain informed consent in CHNS were approved by the institutional review committees of the University of North Carolina at Chapel Hill, as well as the National Institute for Nutrition and Health, Chinese Center for Disease Control and Prevention (201524). All subjects gave written informed consent for their participation in the survey.

## 3. Results

### 3.1. General Information

A total of 15,607 samples from the CHNS 2004, 2006, 2009, 2011, and 2015 waves were involved in this study. The detailed socio-demographic characteristics of the samples are summarized in Table 1.

### 3.2. Dietary Knowledge Trend 2004–2015

Table 2 presents the results of 12 dietary knowledge between 2004 and 2015. Among the 12 dietary knowledge items, the correct rate of 10 items (Q1–Q10) followed upward trends over time from 2004 to 2015 (all *p*-values < 0.05). Compared to 2004, the top three dietary knowledge items with the highest increase in correct rate were Q9 (*consuming beans and bean products is good for one’s health*) (72.8%), Q8 (*consuming milk and dairy products is good for one’s health*) (71.9%), and Q10 (*physical activities are good for one’s health*) (69.2%). However, downward trends were detected for the correct rate of two negative dietary knowledge items, i.e., Q11 (*sweaty sports or other intense physical activities are not good for one’s health*) (*χ*^2^ = 20.05, *p*-value < 0.001) and Q12 (*the heavier one’s body is, the healthier he or she is*) (*χ*^2^ = 43.93, *p*-value < 0.001).

Figure 2 demonstrates the top five dietary knowledge items with the lowest correct rate in 2004–2015. Four of the dietary knowledges consistently followed the lowest correct rate between 2006 and 2015: Q11 (*sweaty sports or other intense physical activities are not good for one’s health*) (24.5%−25.8%), Q5 (*choosing a diet with a lot of staple foods is not good for one’s health*) (36.6%−41.5%), Q6 (*consuming a lot of animal products daily is good for one’s health*) (45.8%−59.5%), and Q7 (*reducing the amount of fatty meat and animal fat in the diet is good for one’s health*) (63.6%−64.9%). In addition, the overall lowest correct rate of dietary knowledge in 2004–2015 was Q12 (*the heavier one’s body is, the healthier he or she is*) (33.6%).

### 3.3. Factors Associated with Dietary Knowledge

#### 3.3.1. Univariate Analysis

A total of 4150 elderly participants were included from the 2015 CHNS to examine the associated factors of dietary knowledge literacy. We found that 30.5% (*n* = 1267) of the participants reported having adequate dietary knowledge literacy. Chi-square tests indicated a significant difference in the dietary knowledge literacy among the participants with different education levels (*χ*^2^ = 130.57, *p* < 0.001), race (*χ*^2^ = 8.90, *p* < 0.01), BMI (*χ*^2^ = 13.81, *p* < 0.01), place of residence (*χ*^2^ = 36.78, *p* < 0.001), geographical area (*χ*^2^ = 197.66, *p* < 0.001), knowing about CFP/DGCR or not (*χ*^2^ = 129.49, *p* < 0.001), and proactively looking for nutrition knowledge or not (*χ*^2^ = 129.49, *p* < 0.001) as shown in Table 3.

#### 3.3.2. Multi-Factor Logistic Regression Analysis

Table 4 demonstrates the results of binary logistic regression analysis for dietary knowledge literacy. When compared to participants who received an education level of below primary school, those who received primary school (*OR* = 1.25, 95% *CI* = 1.02–1.52), middle school (*OR* = 1.37, 95% *CI* = 1.11–1.67), and high school or above (*OR* = 2.07, 95% *CI* = 1.68–2.55) were more likely to have adequate dietary knowledge literacy. Rural participants reported a lower proportion of adequate dietary knowledge literacy than those who lived in urban areas (*OR* = 0.85, 95% *CI* = 0.73–0.99). When compared to participants lived in western regions, those who lived in the northeast (*OR* = 2.85, 95% *CI* = 2.24–3.61), eastern (*OR* = 2.57, 95% *CI* = 2.11–3.14), and central regions (*OR* = 1.39, 95% *CI* = 1.11–1.74) showed better dietary knowledge literacy. In addition, participants who knew about CFP/DGCR (*OR* = 1.71, 95% *CI* = 1.42–2.05) and proactively looked for nutrition knowledge (*OR* = 1.70, 95% *CI* = 1.42–2.03) were more likely to have adequate dietary knowledge literacy than those who did not.

## 4. Discussion

Dietary knowledge plays an important role in developing healthy and reasonable food consumption habits. This study found that the dietary knowledge level of Chinese older adults generally showed upward trends over time between 2004 and 2015, except for two dietary knowledge items regarding physical activity intensity and healthy weight. Elderly people demonstrated a suboptimal dietary knowledge level, such as knowledge related to physical activity intensity and the intake of staple foods, animal products, and fatty meat or animal fat. Educational level, places of residence, geographical region, knowing about CFP/DGCR, and proactively looking for nutrition knowledge were detected to be associated with the dietary knowledge literacy of Chinese older adults.

In this study, the correct rate of most dietary knowledge items (Q1–Q10) increased over time, reflecting the improvement of individual health and health literacy of Chinese older residents. However, there were two exceptions: Q11 (*sweaty sports or other intense physical activities are not good for one’s health*) and Q12 (*the heavier one’s body is, the healthier he or she is*). That is, the knowledge of elderly Chinese residents related to physical activity intensity and healthy weight showed a slight deteriorating trend from 2004 to 2015, which is consistent with Jia et al.’s [18] results among Chinese residents aged 18 years and older. The findings indicated that physical activity intensity and health weight might be two important issues that need special attention in diet-related education. Considering the functional ability of elderly people, they generally tended to and also were recommended to participate in physical activities with low intensity [39,40], which may to some extent explain our finding that older adults believed sweaty physical activities were not beneficial to their fitness. Thus, a specific dietary knowledge list specifically designed for elderly people based on their health characteristics is necessary in China.

We found that the correct rate of four out of the twelve dietary knowledge items consistently ranked top four of the lowest from 2006 to 2015, including two positive items (Q5, *choosing a diet with a lot of staple foods is not good for one’s health*; and Q7, *reducing the amount of fatty meat and animal fat in the diet is good for one’s health*) and two negative items (Q11, *sweaty sports or other intense physical activities are not good for one’s health*; and Q6, *consuming a lot of animal products daily is good for one’s health*).

As for the intake of staple foods, more than half of the participants reported the wrong answer in this study; that is, older adults believed that eating plenty of staple foods was conducive to their health. Wang et al.’s [41] investigation in Jinan, Shandong province reported that half of the rural elderly people exceeded the recommended daily intake of cereals, which supports the present finding. According to the Dietary Guidelines for Chinese Residents 2016, the intake of cereals and tubers for elderly people should be 250–400 g/day, including 50–150 g of whole grains and sweet beans, and 50–100 g of tubers [37]. This suggests that we should change the misunderstanding towards staple food consumption of older residents to promote healthy dietary habits. 

Regarding the intake of animal products and fatty meat and animal fat, elderly people reported insufficient and inaccurate knowledge in this study. Previous studies reported that Chinese older adults tended to eat a large amount of animal products, in particular livestock meat, but their intake of fish and poultry, which were rich in high-quality protein and low in fat, was insufficient [42,43]. According to the DGCR 2016, elderly people are recommended to eat 280–525 g fish, 280–525 g livestock or poultry, and 280–350 g eggs every week, with an average daily intake of 120–200 g [37]. Moynihan et al. [44] found that even if elderly people were aware of the recommendation of reducing fatty food consumption, only approximately one-third of them identified saturated fat as the type of fat to reduce in the diet correctly. These findings suggest the existence of common misunderstandings and a lack of knowledge regarding animal products and fat consumption among Chinese elderly people.

This study indicated that elderly individuals with a higher education level were more likely to have adequate dietary knowledge literacy, which was consistent with Zhang and Xiong’s study [45] among elderly people in nursing homes. Mi et al. [42] proposed that older adults with a higher education level tended to adopt a healthy dietary model, which might be due to a high education level generally representing a higher sense of self-care and awareness of acquiring dietary knowledge. Thus, regarding the popularization of dietary knowledge, older residents with low education levels should be given special attention. Liu et al. [46] reported that people with a low education level were far more likely to acquire nutrition knowledge through television and radio than those with higher education levels. Thus, during the popularizing of dietary knowledge of elderly people, diversified communication channels should be fully utilized, such as offspring or family members, TV channels, and medical practitioners [22].

In this study, better dietary knowledge literacy was detected in elderly participants who lived in urban areas than those who lived in rural areas. Previous studies reported similar results that the dietary knowledge of Chinese older residents exhibited an obvious urban–rural dual structure [23,47]. Accordingly, the elderly’s dietary behaviors also showed significant urban–rural differentiation [48]. During the popularizing of dietary knowledge, it is necessary to identify rural areas as the focus of knowledge publicity. The differences between urban and rural knowledge weak points should be considered. 

A significant association between geographical region and dietary knowledge literacy was observed in the present study. Older adults who lived in eastern, central, and northeast regions were more likely to have adequate dietary knowledge literacy than those who lived in western regions. The proportion of reporting adequate dietary knowledge literacy followed a trend of “western (18.1%) < central (23.3%) < eastern (40.0%) < northeastern (41.1%)” in this study, which was generally in line with previous research [18,29]. However, Kang et al. [10] conducted a multicenter community-based survey in China using the Nutrition Screening Initiative (NSI) and found that elderly residents in eastern regions faced the highest nutritional risks, followed by the western regions, and finally the central regions. These findings suggest that there was great regional inequality in the dietary knowledge and nutritional risk among Chinese older adults, which should be considered in developing targeted diet-related interventions.

Our results indicate that elderly individuals who knew about CFP/DGCR or proactively looked for nutrition knowledge experienced significantly better dietary knowledge literacy than those who did not. Previous studies reported the same findings in Chinese adults aged 18 years and older [18,47]. CFP and DGCR are important guidance for Chinese residents in nutrition. Huang et al. [19] found that elderly residents who knew about CFP/DGCR had a higher daily consumption of fruits, eggs, and milk, as well as higher intake of protein, vitamin B_2_, and calcium, compared to those who did not. 

In 2016, the Chinese Nutrition Society formulated the Dietary Guidelines for the Elderly in China based on the current physiological characteristics, health status, and nutritional needs of the Chinese elderly. The guidelines added relevant contents adapted to the characteristics of the elderly on the basis of the dietary guidelines for the general population. In the future, it is necessary to promote the guidelines in communities and families to improve the awareness rate of the guidelines for all the elderly residents. According to the Nutrition and Health Report of the Elderly in China 2015, nearly one-third of the Chinese elderly were unwilling to accept nutrition knowledge and change their dietary habits for rational nutrition [17]. Therefore, it is necessary to change the nutrition-related attitudes of the elderly by introducing the significance of the appropriate diet before popularizing dietary knowledge.

Several potential limitations should be mentioned regarding the present study. First, all indicators applied in this study were obtained through participants’ self-report, which may cause recall bias due to false or inaccurate responses. Secondly, in terms of the associated factors of dietary knowledge, we used cross-sectional data in 2015 rather than longitudinal data; thus, no causal inferences can be drawn from the results. Thirdly, considering the data timeliness, the 2015 wave of CHNS may not fully reflect China’s current situation. Fourthly, dietary knowledge is not always static and is constantly updated over time. However, due to the limitation of the existing contents of the CHNS questionnaire, this study failed to update the content and standard of dietary knowledge. Despite these limitations, the present study is the first to describe the changing trends of the dietary knowledge of Chinese older adults and to attempt to identify their weak points on dietary knowledge, based on a large sample data. The identification of the associated factors of dietary knowledge provides an important basis for targeted health intervention. These findings might be a valuable reference for the implementation of the current “Appropriate Diet” action and further relevant research.

## 5. Conclusions

In China, older adults demonstrated a suboptimal dietary knowledge level, particularly for knowledge related to physical activity intensity and the intake of staple foods, animal products, and fatty meat or animal fat. The overall correct rates for knowledge regarding physical activity intensity and staple food consumption were only 26.1% and 33.6% between 2004 and 2015. The dietary knowledge level of Chinese older adults generally showed upward trends over time between 2004 to 2015, except for two dietary knowledge items regarding physical activity intensity and healthy weight. Older adults who had a lower educational level, lived in rural areas or western regions, did not know about CFP/DGCR, and did not proactively look for nutrition knowledge were less likely to have adequate dietary knowledge literacy. It is necessary to develop targeted intervention toward the weak points of dietary knowledge among Chinese older residents to improve the healthy level of the elderly. The associated factors above should be considered regarding dietary knowledge interventions for older residents.

## Figures and Tables

**Figure 1 ijerph-17-08029-f001:**
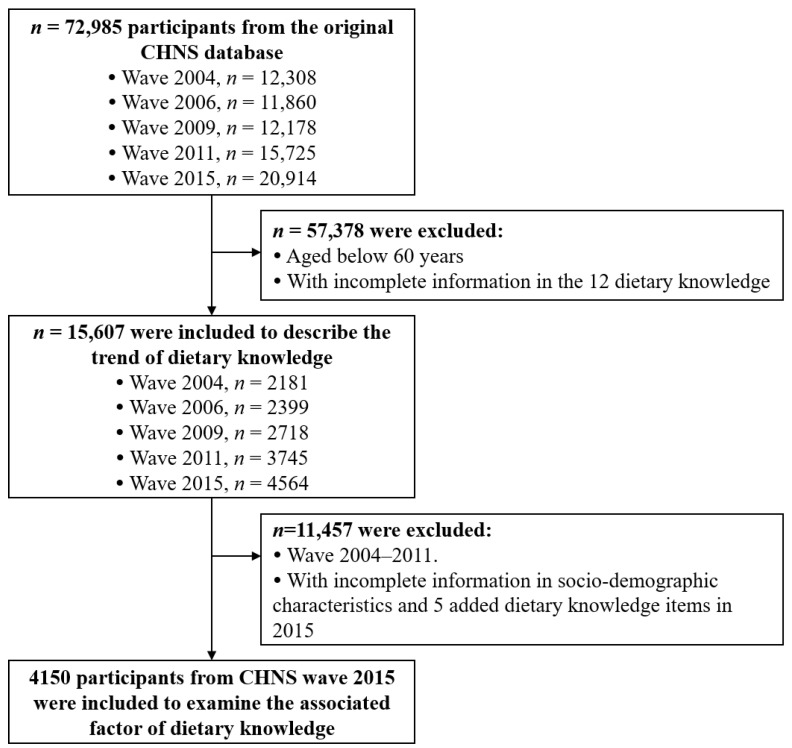
Flow diagram for the sample selection.

**Figure 2 ijerph-17-08029-f002:**
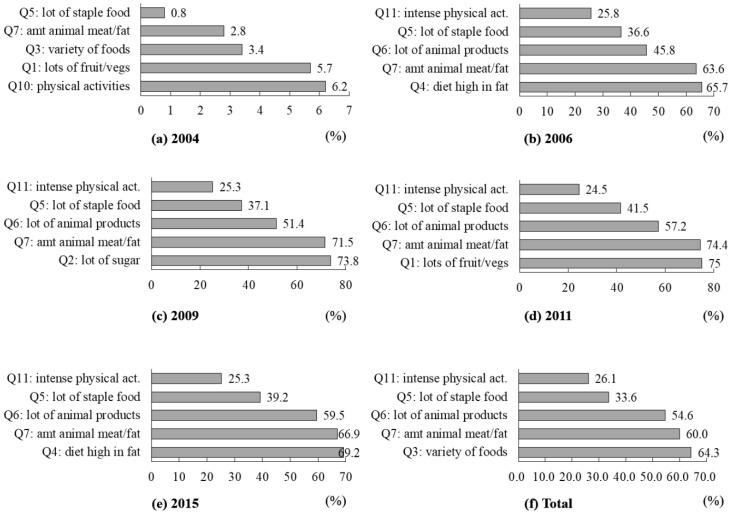
Top five dietary knowledge items with the lowest correct rate in 2004–2015.

**Table 1 ijerph-17-08029-t001:** Participants’ demographic characteristics.

Variables	2004 (*n* = 2181)	2006 (*n* = 2399)	2009 (*n* = 2718)	2011 (*n* = 3745)	2015 (*n* = 4564)
Age (years)					
60–69	1259 (57.7)	1367 (57.0)	1546 (56.9)	2266 (60.5)	2822 (61.8)
70–79	734 (33.7)	815 (34.0)	900 (33.1)	1128 (30.1)	1306 (28.6)
≥80	188 (8.6)	217 (9.0)	272 (10.0)	351 (9.4)	436 (9.6)
Gender					
Male	1024 (47.0)	1124 (46.9)	1275 (46.9)	1775 (47.4)	2136 (46.8)
Female	1157 (53.0)	1275 (53.1)	1443 (53.1)	1970 (52.6)	2428 (53.2)
Education level					
Below primary school ^a^	1116 (51.4)	1297 (54.4)	1324 (48.9)	1573 (42.1)	1497 (32.8)
Primary school	525 (24.2)	451 (18.9)	633 (23.4)	795 (21.3)	1043 (22.9)
Middle school	260 (12.0)	307 (12.9)	401 (14.8)	716 (19.2)	1044 (22.9)
High school or above	272 (12.5)	331 (13.9)	351 (13.0)	651 (17.4)	980 (21.5)
Nationality					
Han	1933 (88.6)	2109 (87.9)	2351 (86.8)	3357 (89.8)	4078 (90.1)
The minority	248 (11.4)	290 (12.1)	358 (13.2)	381 (10.2)	448 (9.9)
BMI (kg/m^2^)					
<20	444 (21.5)	476 (21.2)	494 (19.1)	571 (15.6)	493 (11.8)
20–26.9	1318 (63.9)	1452 (64.7)	1710 (66.0)	2463 (67.1)	2825 (67.5)
≥27	300 (14.5)	317 (14.1)	386 (14.9)	637 (17.4)	868 (20.7)
Place of residence					
Urban	839 (38.5)	892 (37.2)	979 (36.0)	1599 (42.7)	1915 (42.0)
Rural	1342 (61.5)	1507 (62.8)	1739 (64.0)	2146 (57.3)	2649 (58.0)
Geographical region					
Western region	596 (27.1)	650 (27.1)	730 (26.9)	1011 (27.0)	1218 (26.7)
Central region	665 (30.4)	729 (30.8)	837 (30.8)	939 (25.1)	1068 (23.4)
Eastern region	554 (25.4)	609 (25.4)	681 (25.1)	1279 (34.2)	1680 (36.8)
Northeast region	366 (16.8)	411 (17.1)	470 (17.3)	516 (13.8)	598 (13.1)

Sample sizes of the demographic characteristics variables may not sum to *n* = 15,607 due to missing values. ^a^ Below primary school group includes participants who were uneducated or did not know their own education level. BMI, body mass index.

**Table 2 ijerph-17-08029-t002:** The proportion of correctly answering on 12 dietary knowledge in 2004–2015.

		2004	2006	2009	2011	2015	Cochran-Armitage *χ*^2^
Q1	Choosing a diet with a lot of fresh fruits and vegetables is good for one’s health.	124 (5.7)	1799 (75.0)	2031 (74.7)	2807 (75.0)	3393 (74.3)	1515.41 ^a^ ***
Q2 ^c^	Eating a lot of sugar is good for one’s health.	1536 (70.4)	1631 (68.0)	2007 (73.8)	2952 (78.8)	3222 (70.6)	4.65 ^a^ *
Q3	Eating a variety of foods is good for one’s health.	74 (3.4)	1794 (74.8)	2005 (73.8)	2861 (76.4)	3303 (72.4)	1487.21 ^a^ ***
Q4 ^c^	Choosing a diet high in fat is good for one’s health.	1510 (69.2)	1577 (65.7)	2013 (74.1)	2888 (77.1)	3158 (69.2)	4.46 ^a^ *
Q5	Choosing a diet with a lot of staple foods (rice and rice products, wheat and wheat products) is not good for one’s health.	17 (0.8)	879 (36.6)	1009 (37.1)	1554 (41.5)	1790 (39.2)	567.81 ^a^ ***
Q6 ^c^	Consuming a lot of animal products daily (fish, poultry, eggs and lean meat) is good for one’s health.	1160 (53.2)	1099 (45.8)	1398 (51.4)	2141 (57.2)	2717 (59.5)	92.75 ^a^ ***
Q7	Reducing the amount of fatty meat and animal fat in the diet is good for one’s health.	62 (2.8)	1525 (63.6)	1943 (71.5)	2786 (74.4)	3054 (66.9)	1403.74 ^a^ ***
Q8	Consuming milk and dairy products is good for one’s health.	149 (6.8)	1959 (81.7)	2258 (83.1)	3195 (85.3)	3593 (78.7)	1801.89 ^a^ ***
Q9	Consuming beans and bean products is good for one’s health.	169 (7.7)	2046 (85.3)	2347 (86.4)	3235 (86.4)	3672 (80.5)	1827.50 ^a^ ***
Q10	Physical activities are good for one’s health.	136 (6.2)	1972 (82.2)	2198 (80.9)	3120 (83.3)	3440 (75.4)	1475.46 ^a^ ***
Q11 ^c^	Sweaty sports or other intense physical activities are not good for one’s health.	698 (32.0)	619 (25.8)	689 (25.3)	919 (24.5)	1156 (25.3)	20.05 ^b^ ***
Q12 ^c^	The heavier one’s body is, the healthier he or she is.	1706 (78.2)	1960 (81.7)	2227 (81.9)	3222 (86.6)	3343 (73.2)	43.93 ^b^ ***

Q, question. ^a^ upward trend; ^b^ downward trend; ^c^ negative item. * *p* <0.05; ** *p* <0.01; *** *p* <0.001.

**Table 3 ijerph-17-08029-t003:** The dietary knowledge literacy of participants with different demographic characteristics in 2015.

Variables	Categories	Total	Adequate Dietary Knowledge Literacy		*χ* ^2^
			Yes (*n* = 1267, 30.5%)	No (*n* = 2883, 69.5%)	
Age (years)	60–69	2630	831 (31.6)	1799 (68.4)	4.63
	70–79	1194	336 (28.1)	858 (71.9)	
	≥80	326	100 (30.7)	226 (69.3)	
Gender	Female	1932	617 (31.9)	1315 (68.1)	3.37
	Male	2218	650 (29.3)	1568 (70.7)	
Education level ^a^	Below primary school	1334	295 (22.1)	1039 (77.9)	130.57 ***
	Primary school	970	272 (28.0)	698 (72.0)	
	Middle school	960	305 (31.8)	655 (68.2)	
	High school or above	886	395 (44.6)	491 (55.4)	
Nationality	The minority	400	96 (24.0)	304 (76.0)	8.90 **
	Han	3750	1171 (31.2)	2579 (68.8)	
BMI (kg/m^2^)	<20	491	121 (24.6)	370 (75.4)	13.81 **
	20–26.9	2799	851 (30.4)	1948 (69.6)	
	≥27	860	295 (34.3)	565 (65.7)	
Place of residence	Urban	1724	615 (35.7)	1109 (64.3)	36.78 ***
	Rural	2426	652 (26.9)	1774 (73.1)	
Geographical area	Western region	1099	199 (18.1)	900 (81.9)	197.66 ***
	Eastern region	1580	632 (40.0)	948 (60.0)	
	Central region	948	221 (23.3)	727 (76.7)	
	Northeast region	523	215 (41.1)	308 (58.9)	
Knowing about CFP/DGCR	No	3166	823 (26.0)	2343 (74.0)	129.49 ***
	Yes	984	444 (45.1)	540 (54.9)	
Proactively looking for nutrition knowledge	No	3081	793 (25.7)	2288 (74.3)	129.49 ***
	Yes	1069	474 (44.3)	595 (55.7)	

* *p* < 0.05; ** *p* < 0.01; *** *p* < 0.001. BMI, body mass index; CFP, Chinese Food Pagoda; DGCR, Dietary Guidelines for Chinese Residents.

**Table 4 ijerph-17-08029-t004:** Multi-factor logistic regression analysis testing the factors associated with dietary knowledge literacy.

Variables	*OR*	95% *CI* for *OR*		*p*-Value
		Lower	Upper	
Education level (vs. Below primary school)				
Primary school	1.25	1.02	1.52	0.029
Middle school	1.37	1.11	1.67	0.003
High school or above	2.07	1.68	2.55	0.000
Race (vs. The minority)				
Han	0.93	0.72	1.21	0.590
BMI (vs. <20)				
20–26.9	1.07	0.85	1.35	0.570
≥27	1.22	0.94	1.58	0.142
Place of residence (vs. Urban areas)				
Rural areas	0.85	0.73	0.99	0.033
Geographical area (vs. Western region)				
Eastern region	2.57	2.11	3.14	0.000
Central region	1.39	1.11	1.74	0.004
Northeast region	2.85	2.24	3.61	0.000
Knowing about CFP/DGCR (vs. No)				
Yes	1.37	1.13	1.67	0.002
Proactively looking for nutrition knowledge (vs. No)				
Yes	1.39	1.15	1.68	0.001

Goodness-of-fit test: Hosmer–Lemeshow test *χ*^2^ = 11.88, *p*-value = 0.156. *OR*, odds ratio; *CI*, confidence interval; BMI, body mass index; CFP, Chinese Food Pagoda; DGCR, Dietary Guidelines for Chinese Residents.

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
