# Peer review of "Trends and Associated Factors of Dietary Knowledge among Chinese Older Residents: Results from the China Health and Nutrition Survey 2004–2015"

_ijerph, 2020, doi:10.3390/ijerph17218029_

Round 1

Reviewer 1 Report

This is an interesting paper on dietary knowledge among Chinese older residents using the China Health and Nutrition Survey data. The reviewer has no major comment, and only minor comments are shown below.

1. Please describe the procedure how the authors acquired the CHNS data.
2. Please cite the references on BMI criteria (L110), Chinese Food Pagoda (L117) and Dietary Guidelines for Chinese Residents (L117).
3. Sentences on the written informed consent are duplicated (L147-L149).
4. In Table 3, the percentages are confusing because those for the total are for the column and those for the adequate dietary knowledge literacy are for the row. Those for the total can be deleted.

Reviewer 2 Report

In my opinion, this is an interesting study and has value in promoting the health of elderly. The lack of correct dietary knowledge among the elderly in China is a widespread problem which has been ignored for a long time.It is a quite significant study with some good insights, but also with some points need to be revised.

Firstly, the definition of the so-called "correct dietary knowledge" is not clear enough. In fact, dietary knowledge will be constantly updated over time, and the authors should seek a relatively stable standard for what is "correct dietary knowledge". Moreover, I recommend careful editing to reduce the focus on common sense and emphasize new content.

secondly, there are some minor errors in the text, which may affect the readability and review of the article. For example, "Shaanxi" on line 86 may be "Shanxi", there is a lack of comma before "Q6" in line 174, and "the" in line 176 should be "the", etc. This requires further attention. Such minor errors should be thoroughly and carefully checked when revising.

Finally, the manuscript requires some editing to render it suitable for publication in an English language journal.
